# A Systematic Review on Processed/Ultra-Processed Foods and Arterial Hypertension in Adults and Older People

**DOI:** 10.3390/nu14061215

**Published:** 2022-03-13

**Authors:** Suamy Sales Barbosa, Layanne Cristini Martin Sousa, David Franciole de Oliveira Silva, Jéssica Bastos Pimentel, Karine Cavalcanti Maurício de Sena Evangelista, Clélia de Oliveira Lyra, Márcia Marília Gomes Dantas Lopes, Severina Carla Vieira Cunha Lima

**Affiliations:** 1Post Graduate Program in Nutrition—Department of Nutrition, Federal University of Rio Grande do Norte, Natal 59078-970, RN, Brazil; jessicapimentel@ufrn.edu.br (J.B.P.); karine.sena@ufrn.br (K.C.M.d.S.E.); clelia.lyra@ufrn.br (C.d.O.L.); marilia.lopes@ufrn.br (M.M.G.D.L.); severina.lima@ufrn.br (S.C.V.C.L.); 2Department of Nutrition, Federal University of Rio Grande do Norte, Natal 59078-970, RN, Brazil; 3Post Graduate Program in Collective Health—Health Sciences Center, Federal University of Rio Grande do Norte, Natal 59056-000, RN, Brazil; layanne.sousa.087@ufrn.edu.br (L.C.M.S.); davfranci@hotmail.com (D.F.d.O.S.)

**Keywords:** hypertension, blood pressure, dietary habits, food processing, NOVA classification, systematic review, chronic non-communicable diseases

## Abstract

The increase in the availability of processed and ultra-processed foods has altered the eating patterns of populations, and these foods constitute an exposure factor for the development of arterial hypertension. This systematic review analyzed evidence of the association between consumption of processed/ultra-processed foods and arterial hypertension in adults and older people. Electronic searches for relevant articles were performed in the PUBMED, EMBASE and LILACS databases. The review was conducted following the PRISMA guidelines and the Newcastle–Ottawa Scale. The search of the databases led to the retrieval of 2323 articles, eight of which were included in the review. A positive association was found between the consumption of ultra-processed foods and blood pressure/arterial hypertension, whereas insufficient evidence was found for the association between the consumption of processed foods and arterial hypertension. The results reveal the high consumption of ultra-processed foods in developed and middle-income countries, warning of the health risks of such foods, which have a high energy density and are rich in salt, sugar and fat. The findings underscore the urgent need for the adoption of measures that exert a positive impact on the quality of life of populations, especially those at greater risk, such as adults and older people.

## 1. Introduction

Ultra-processed foods (UPFs) are industrial formulations and constitute an exposure factor for the development of arterial hypertension (AH) [1,2,3], which is considered the main risk factor for the major cause of mortality throughout the world—cardiovascular disease [4]. The term “ultra-processed” corresponds to one of the four classifications of the NOVA system, which groups foods according to the degree of processing—in natura or minimally processed, cooking ingredients, processed foods (PFs) and UPFs. The UPFs are hypercaloric and have an unbalanced nutritional composition. Such foods have attractive organoleptic characteristics (high palatability and colorful) and are inexpensive, but constitute a risk to human health. Examples include packaged chips, snacks, soft drinks, artificial juices, cookies and frozen/pre-prepared meals [5,6].

The PFs also merit attention. These foods are essentially manufactured with the addition of salt or sugar to an in natura or minimally processed food, such as canned vegetables, fruit in syrup, cheeses and some types of bread [5,6]. The increase in the availability of PFs and UPFs and the simultaneous occurrence of the nutritional transition have altered the eating pattern of populations. Traditional cooking and eating habits based on in natura and minimally processed foods have largely been replaced by convenient PF/UPFs, increasing the risk of the development of diseases [7,8]. Studies report an association between food components such as sodium and alcohol as a risk factor, whereas potassium, magnesium and calcium offer protection from the development of AH [9,10,11,12,13]. To the best of our knowledge, however, few studies have evaluated the impact of the consumption of PFs and UPFs considering AH as the outcome.

From the public health standpoint, it is important to assess the severity and magnitude of AH and its association with the increase in the consumption of PFs and UPFs [8,14]. Changes in eating patterns and lifestyle in populations throughout the world in recent decades underscore the urgent need for interventions on the part of governments to address the increase in the prevalence of AH. Therefore, the aim of the present study was to analyze evidence of the association between the consumption of PFs/UPFs and AH in adults and older people.

## 2. Materials and Methods

This study was conducted following the Preferred Reporting Items for Systematic Reviews and Meta-Analyses (PRISMA) guidelines 2020 [15]. All information on the search, article selection process and data extraction were previously registered with the International Prospective Register of Systematic Reviews (PROSPERO—CRD42021222514). The following was the guiding question: “Is there an association between processed/ultra-processed foods and arterial hypertension in adults and older people?”

The PICOS acronym was used in the design of the study for the definition of the inclusion and exclusion criteria: P (Population)—adults (20 to 59 years of age) and/or older people (60 years of age or older); I (Intervention/Exposure)—high consumption of processed and ultra-processed foods based on the NOVA classification; C (Comparison)—low consumption of processed and ultra-processed foods based on the NOVA classification; O (Outcome)—arterial hypertension defined based on any diagnostic criteria; S (Type of Study)—observational (cohort, case-control and cross-sectional) and intervention studies (Appendix A).

Studies that satisfied the criteria established using the PICOS method and evaluated the consumption of PFs and/or UPFs and its association with AH in adults and/or older people were included. No restrictions were imposed with regards to language or year of publication. Studies with pregnant women, children and adolescents, those that addressed a disease other than AH, review articles, guidelines, letters and editorials were excluded. Studies that used the terms “processed” or “ultra-processed” but did not follow the requirements of the NOVA classification proposed by Monteiro et al. (2010) [5] were not included.

### 2.1. Search Strategy/Inclusion and Exclusion Criteria

Two reviewers (S.S.B. and L.C.M.S.) performed independent searches of the PubMed, Embase and LILACS databases on 20 May 2021. For the EMBASE database, the search was refined by selecting only articles and articles in press among the different publication types. The following search strategies were employed:-PubMed: (“ultra-processed food” OR “ultra-processed foods” OR “ultraprocessed food” OR “ultraprocessed foods” OR “ultra-processed product” OR “ultra-processed products” OR “ultra-processing” OR “food processing” OR “processed food” OR “processed foods” OR “NOVA” OR “NOVA system” OR “NOVA food classification” OR “NOVA classification system”) AND (hypertension OR “high blood pressure” OR “high blood pressures” OR “blood pressure” OR “systolic pressure” OR “diastolic pressure” OR “systolic blood pressure” OR “diastolic blood pressure”) AND (adult OR adults OR aged OR “middle aged” OR elderly OR “older adult”).-Embase: (“ultra-processed food” OR “ultra-processed foods” OR “ultraprocessed food” OR “ultraprocessed foods” OR “ultra-processed product” OR “ultra-processed products” OR “ultra-processing” OR “food processing” OR “processed food” OR “processed foods” OR “NOVA” OR “NOVA system” OR “NOVA food classification” OR “NOVA classification system”) AND (hypertension OR “high blood pressure” OR “high blood pressures” OR “blood pressure” OR “systolic pressure” OR “diastolic pressure” OR “systolic blood pressure” OR “diastolic blood pressure”) AND (adult OR adults OR aged OR “middle aged” OR elderly OR “older adult”).-LILACS: (“alimento ultra-processado” OR “alimentos ultra-processados” OR “alimento ultraprocessado” OR “alimentos ultraprocessados” OR “produto ultra-processado” OR “produtos ultra-processados” OR “ultra-processamento” OR “processamento de alimento” OR “alimento processado” OR “alimentos processados” OR “NOVA” OR “sistema NOVA” OR “classificação de alimentos NOVA” OR “sistema de classificação de alimentos NOVA”) AND (hipertensão OR “hipertensão arterial sistêmica” OR “pressão arterial elevada” OR “pressão arterial” OR “pressão sistólica” OR “pressão diastólica” OR “pressão arterial sistólica” OR “pressão arterial diastólica”) AND (adulto OR adultos OR idoso OR idosos).

### 2.2. Article Selection Process and Data Extraction

Articles were retrieved from the databases using the search terms. Duplicates were removed and the selection process for the review was conducted in two steps. The titles and abstracts were analyzed for the preselection of potentially eligible articles and the exclusion of those that did not meet the objectives of the review. The preselected articles were then submitted to full-text analysis for the selection of those that met the inclusion criteria. The articles selected by each reviewer were compared. In cases of a divergence of opinion, a third reviewer (D.F.d.O.S.) was consulted to make the decision regarding inclusion or exclusion.

The following data were extracted from the articles selected for the present review: author and year of publication, country in which the study was conducted, language in which the article was published, sample size, age of participants, food consumption assessment method, denomination and composition of dietary components, method used in the statistical analysis, criteria for the diagnosis of AH, energy contribution of PFs/UPFs and results of associations between processing of foods and AH.

### 2.3. Appraisal of Methodological Quality

The appraisal of methodological quality and risk of bias in the cohort and cross-sectional studies was performed using the Newcastle–Ottawa scale [16] and adapted Newcastle–Ottawa scale [17], respectively. The articles were classified as “poor”, “fair”, “good” or “excellent” when achieving scores of 0–3, >3–6, >6–8 or >8–9, respectively.

Synthesis Without Meta-analysis (SWiM) [18] was used for the narrative description of the data on the association between the consumption of PFs/UPFs and AH in adults and older people. The number of studies with a positive association between PFs/UPFs and AH was compared to the number reporting an inverse association and those reporting no association to determine the summary of the evidence.

## 3. Results

### 3.1. Article Selection Process

The search of the databases led to the retrieval of 2323 articles: 741 in PubMed, 1486 in Embase and 97 in LILACS. After the removal of duplicates, the articles were screened based on the reading of the title and abstract. Review articles, non-observational studies, those that evaluated outcomes other than AH, those with samples of children, adolescents or pregnant women and animal studies were excluded. Ten potentially eligible articles were submitted to full-text analysis, nine of which were included in the present review. Figure 1 shows the flowchart of the article selection process.

### 3.2. Overview and Characteristics of Studies

The nine articles selected were published in the last five years (2017 to 2021) and were conducted in seven countries of the Americas and two in Europe: Brazil (*n* = 3) [2,19,20], USA (*n* = 2) [21,22], Canada (*n* = 1) [3], Mexico (*n* = 1) [23] and Spain (*n* = 2) [1,24]. The objective of the studies and dietary component analyzed are displayed in Table 1.

Six studies had samples composed only of adults [1,2,3,19,21,23] and three had sample composed of adults and older people [20,22,24]. One study only evaluated women [23]. A total of 114,849 individuals participated in the nine studies. Five of the studies had a cross-sectional design [3,19,21,22,24] and four were cohort studies [1,2,20,23] (Table 2). The average duration of the cohort studies was 3.5 years.

### 3.3. Processed and Ultra-Processed Food Consumption

The predominant data collection tool for the food consumption assessment in the different populations was the food frequency questionnaire (FFQ) (*n* = 5), followed by the 24-h recall (*n* = 3) and food record (*n* = 1). All studies used the NOVA classification proposed by Monteiro et al., (2010) [5] to categorize the foods based on the degree of processing. For the present review, only analysis performed with foods from the processed and ultra-processed categories were considered. Seven studies exclusively analyzed UPFs and two analyzed both PFs and UPFs. No studies analyzed PFs alone. The average daily caloric contribution ranged from 6.2 to 9.9% for PFs and from 7.7% [19] to 55.5% [22] for UPFs. American and Canadian populations had the highest consumption of UPFs.

### 3.4. Association between Processing of Food and Arterial Hypertension

Seven studies included in the present review strictly analyzed the association between the consumption of PFs and/or UPFs and BP and/or AH [1,2,19,20,21,23,24]. Two studies analyzed the association between the consumption of these foods and metabolic syndrome [22], obesity, diabetes, AH and heart disease [3]. Both studies were included because BP/AH was one of the variables evaluated in relation to UPFs. The effect measures used to determine the association between the consumption of PFs/UPFs and AH were the odds ratio (OR), hazard ratio (HR), incidence rate ratio (IRR) and risk ratio (RR) with respective 95% confidence intervals (CI). The types of statistical analysis used in each study are described in Table 2.

Among the articles analyzed, eight used diverse covariables in the multivariate analyses. However, only five studiesowever, only Ho [1,2,20,22,24] included biochemical data as adjustment variables in the regression models. Among these studies, one [22] analyzed the association between UPFs and metabolic syndrome and not specifically the association between UPFs and AH/BP. Steele et al. [22] found a significant association between the quintiles of UPF consumption and high BP.

Nearly all studies (*n* = 7) found a positive association between the consumption of PFs/UPFs and AH/BP. Only two cross-sectional studies [19,24] found no statistically significant difference in the average SBP and DBP based on the consumption of these foods.

### 3.5. Quality Appraisal

The complete appraisal of the methodological quality of the articles is described in Appendix A. The cross-sectional studies had scores ranging from 4 to 7 and the cohort studies had scores ranging from 6 to 7.

## 4. Discussion

The present systematic review found a positive association between the consumption of UPFs and BP/AH, pointing out the health risk of the consumption of highly PFs, which have a high energy density and are rich in salt, sugar and fat. Previous reviews have also evaluated the effect of these foods on different health outcomes, such as cardiovascular disease, cerebrovascular disease, overweight, obesity, depression and metabolic syndrome [31,32,33,34,35]. Such findings offer evidence that the consumption of these foods has negative consequences for human health.

The sample sizes in the articles of the present review ensure representativity and confer reliability to the results, suggesting adequate quality of the evidence presented. Only two cross-sectional studies [19,24] found no statistically significant difference in the average SBP and DBP based on the consumption of UPFs. In one of the articles, although the methodological quality of the study was considered satisfactory, the sample size was relatively small (64 participants) [19]. On the other [24], the consumption of UPFs was significantly associated only with anthropometric data (weight, BMI and waist circumference) and biochemical data (HDL and creatinine). However, there is a consensus that overweight, obesity, excess abdominal fat and low HDL are cardiometabolic risk factors and caution should be exercised when consuming UPFs [36,37].

Most of the participants in the studies included in the present review had a higher education (more than 90,000 individuals). In the Brazilian study conducted by Scaranni et al. [20], in which 58% of the sample had a university degree, a higher level of schooling was associated with a greater consumption of UPFs. Three explanations may be offered for these findings: (1) individuals with higher education may have less time available to prepare meals due to their academic and professional activities; (2) the possibility of a higher income in this population implies greater freedom in food choices, with the acquisition of inadequate foods; (3) UPFs are products that may meet the needs of this population in terms of practicality, variety and convenience, constituting an exposure factor for AH. The influence of schooling and income level on diet indicates the need to evaluate different groups in population-based studies for a more precise identification of eating patterns.

Most of the articles included in this review were of population-based studies with representative samples and the investigation of different variables. Thus, the authors sought to investigate other possible factors (sociodemographic characteristics, lifestyle, health conditions, etc.) correlated with the consumption of UPFs and AH. For these types of analyses, multiple regression models were used to control for the effect of confounding variables in the associations. One study used the Student’s t-test as the analysis method [19]. The statistical analyses employed in the studies were adequate to the design and objective, making the results more consistent.

The use of adequate data collection tools for the determination of food intake ensures greater reliability of the results. Moreover, it is important for the instruments used to be validated specifically for the objectives and population one wishes to evaluate, such as the assessment tool developed by Mota et al. [38], which enhances the quality of the evidence [39]. In the present review, five studies used a validated FFQ [1,2,20,23,24] for the populations studied, but without validation for the assessment of food intake according to the degree of processing. This factor increases the likelihood of the underestimation or overestimation of the consumption of PFs/UPFs. In a previous systematic review, Marino et al. [33] found that most data on food intake were from FFQs not validated for estimating UPFs and, therefore, the results should be interpreted with caution, especially when used to analyze associations with health status.

Two cross-sectional studies found associations between the high consumption of UPFs in American [21] and Canadian [3] populations and the development of AH. The UPFs have dominated the food supply in high-income countries and the consumption of these foods is rapidly increasing in middle-income countries [40]. Studies with national representativeness conducted in Canada and the USA have identified changes in eating patterns [28,30]. In Canada, whole or minimally PFs and cooking ingredients have been replaced with ready-to-eat meals and other UPFs [28]. A study conducted in the USA found that the consumption of UPFs accounted for 57.9% of the calorie intake of Americans [30].

The cohort studies included in the present review conducted in Brazil also found a greater risk of the development of AH among individuals who consumed more UPFs, and the study conducted in Mexico found an association between subgroups of UPFs (meats and beverages) and an increase in the incidence of AH [2,20,23]. These associations reflect an increase in urban living and the influence of foreign markets on the Latin American economy [41]. A study on the risk of the development of cardiovascular disease in middle-aged Americans demonstrated that the prevalence of AH was higher among those who consumed larger quantities of UPFs [42]. These data underscore the need to investigate the eating habits of populations and associated factors and establish strategies to attenuate the negative consequences of the excessive consumption of these foods with regards to BP.

For the assessment of AH or altered BP, most of the cross-sectional studies measured BP at the time of data collection [19,21,22,24]. Only one study obtained this information based on a self-declared medical diagnosis during the interview [3]. The cohort studies collected these data through questionnaires sent to the participants during the follow-up period. Some considered AH only in the occurrence of a self-declaration of a medical diagnosis of AH [1], and others considered AH in the occurrence a self-declared high BP or the use of an antihypertensive in a particular period of time [2,23]. To minimize errors during the gathering of information, the cohort studies previously validated the tool used for the collection of BP data. Only one cohort study involved the measurement of BP throughout the follow-up period and also obtained information on the use of antihypertensives [20].

Few studies evaluated the consumption of PFs [2,19] and it was, therefore, not possible to measure the impact of the consumption of these foods on BP or the development of AH. The majority of studies investigated the association between the consumption of UPFs and AH. Ultra-processing poses a public health challenge, as such foods have the advantages of being inexpensive, highly palatable and convenient and have a long shelf-life, but are characteristically energy dense and have high contents of fat, sugar and salt. Moreover, the formulation, packaging and marketing often induce excessive consumption [25]. Ultra-processing leads to the production of unhealthy foods that are rich in energy and poor in protective micronutrients and fiber, resulting in “empty calories” [43]. Thus, there is a need for large-scale strategies and public policies that involve all actors participating in this process—from the production chain (through the promotion of agroecology and sustainable food systems) to the consumer—in order to reduce the consumption of UPFs and encourage healthier eating habits.

The Food and Agriculture Organization established goals to ensure sustainable consumption and production patterns (12th Sustainable Development Goal: Responsible Consumption and Production). For future food systems to be sustainable, new (or forgotten) ideas, practices and forms of organization are needed to ensure that all activities that bring foods grown in soil or aquatic organisms to the table of populations are environmentally sustainable as well as economically inclusive and socially fair. It is, therefore, fundamental to seek joint strategies for the protection of the health of populations and intervene on local, regional, national and international levels with the participation of civil society, researchers, governments and the private sector [44].

The innovative NOVA food classification method based on the degree of processing [5] has altered the interpretation of what constitutes healthy foods and the repercussions with regards to health and the development of diseases. The notion that foods should be analyzed in their totality, encompassing the content of nutrients and ingredients, highlights the importance of considering all aspects from the beginning of the food production process to the consumer’s table. This perspective poses challenges for the creation of new assessment tools and investigation methods that identify the impact of the degree of food processing on the health of the population.

Despite studies in the literature involving the NOVA classification, analyses on the association between PFs/UPFs and AH are scarce, especially those involving adults and older people. In the present review, only nine such studies were found, most of which were conducted in the Americas. To the best of our knowledge, no studies of this type have been conducted with Asian or African communities. Thus, there is a gap in knowledge to be filled with further studies. Among the scientific publications, we found a diversity of studies involving samples of children [45], adolescents [46] and pregnant women [47], as well as those that investigated the association between UPFs and the occurrence of obesity/weight gain [48], metabolic syndrome and cardiovascular disease. Studies with an adequate, robust methodological design for the determination of the cause-and-effect relationship, such as randomized clinical trials, involving representative populations on different continents, could further clarify the impact of PFs/UPFs on BP and the development of AH, especially in populations at greater risk, such as adults and older people.

### Limitations and Strengths 

The present review has limitations that should be considered. First, studies with different methodological designs (cross-sectional and cohort) were included. This decision was made due to the scarcity of studies investigating the association between PFs/UPFs and BP/AH in adults and older people. Second, some of the studies used a food frequency questionnaire not specifically validated for the collection of data on food intake according to the NOVA classification, which may have resulted in the underestimation or overestimation of the consumption of PFs/UPFs. Third, few studies were found that evaluated the consumption of PFs and involved the older population, possibly due to the fact that PFs are not considered to be as harmful as UPFs and that more discerning methodological criteria are needed for the assessment of older people.

This review also has strong points, such as the originality of the study in terms of the investigation of the association between the consumption of PFs/UPFs and BP/AH. To the best of our knowledge, this review is a pioneering study on this subject. Secondly, the review presents data on the main risk factor for cardiovascular disease and, consequently, the main cause of morbidity and mortality throughout the world—hypertension. Lastly, a rigorous selection of articles was performed according to predetermined inclusion criteria, with the inclusion only of studies in which the classification of PFs/UPFs faithfully followed the characteristics proposed by the NOVA system.

## 5. Conclusions

Based on the findings of the present review, UPFs are associated with a greater risk of developing AH in the adult population and older people. The evidence underscores the need to investigate the eating habits of populations due to the increase in the consumption of unhealthy foods, which can have negative health consequences. Such knowledge could assist in the adoption of measures that have a positive impact on the transformation of the health scenario in the long term.

## Figures and Tables

**Figure 1 nutrients-14-01215-f001:**
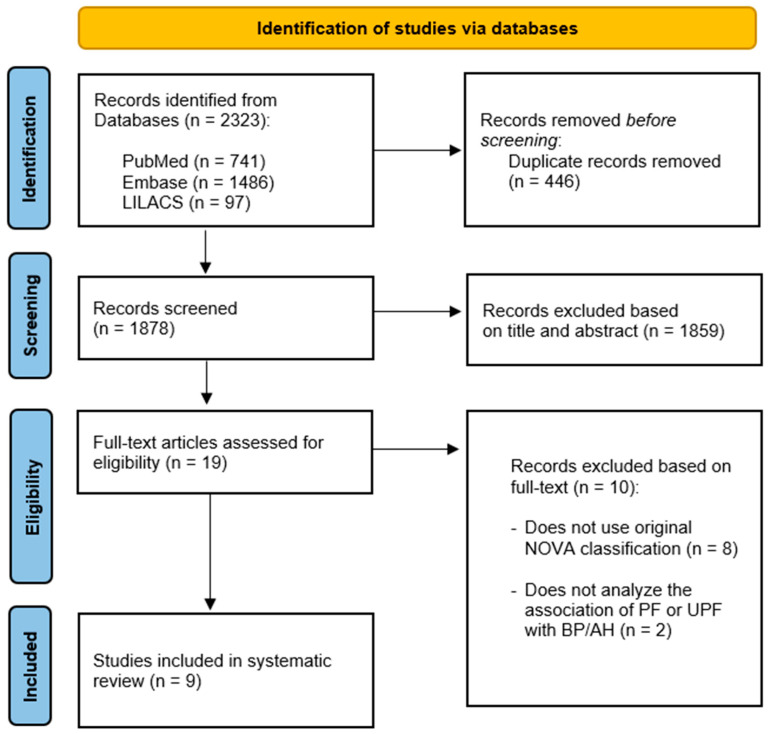
PRISMA flowchart of the included studies.

**Table 1 nutrients-14-01215-t001:** Overview of studies selected for present review (*n* = 9).

Author (Year)	Language of Publication	Objective of Study	Denomination and Description of Dietary Component Evaluated
Conceição et al., (2018) [19]	English	Evaluate whether intake of macronutrients and micronutrients and blood pressure (BP) levels are associated with degree of food processing	PFs: Salt, sugar or other substance of culinary use added to the food in natura or minimally processed (roasted biscuit; mozzarella; salted bread; whole grain bread; Minas cheese; toast). UPFs: essentially industrial food, ready to eat, multi-ingredient products involving multiple steps and processing techniques (chocolate; normal and whole grain salt and water crackers; corn starch and polvilho crackers; pasta with tomato sauce; margarine; light margarine; instant powder for porridge; cream cheese; salami; artificial strawberry and grape juices.
Martinez-Peres et al., (2021) [24]	English	Assess the impact of the food classification system on the association between the consumption of UPFs and cardiometabolic health using the same dataset.	UPFs: article followed description proposed by Monteiro et al., (2018) [25], Monteiro et al., (2011) [26], Monteiro et al., (2016) [27].
Mendonça et al., (2017) [1]	English	Evaluate potential association between consumption of UPFs and risk of AH	AUPs: carbonated drinks, processed meat, biscuits, cookies, candy, confectionery, ‘instant’ packaged soups and noodles, sweet or savory packaged snacks, and sugared milk and fruit drinks. Article followed description proposed by Monteiro et al., (2010), Monteiro et al., (2016), Moubarac et al., (2014) [5,27,28].
Monge et al., (2021) [23]	English	Estimate association between incidence of AH and consumption of UPFs (liquids and solids) as well as subgroups of UPFs	UPFs: industrial formulations with multiple ingredients that are usually not used for cooking (like food additives), such as sugar-sweetened beverages (SSB), packed snacks and candies. The UPFs were classified into subgroups dairy products (yogurt, ice cream, petite suisse, Yakult), added fats (cream, margarine, cream cheese), sugary products (jello, flan, sweet breads, cakes, cookies, candies, chocolate, honey, jelly and fruit paste candy), SSB (soya milk, orange juice, soda, flavored water), alcoholic beverages, processed meats (bacon, sausage, ham, chorizo, longaniza (a spicy pork sausage) and other deli meats), cereals (processed oats, low- and high-fiber breakfast cereals, cereal bars, white and whole-grain loaf of bread), salty snacks (chips and saltines) and fast food (burgers, hotdogs, pizza, tortas).
Nardocci et al., (2020) [3]	English	Evaluate associations between consumption of UPFs and obesity, diabetes, AH and heart disease	UPFs: article followed description proposed by Moubarac et al., (2017) for UPFs [29].
Rezende-Alves el at., (2020) [2]	English	Analyze association between consumption of foods according to degree of processing and incidence of AH	PFs and UPFs: complete list of PFs and UPFs in supplementary material of article by Rezende-Alves et al., (2020) based on description proposed by Monteiro et al., (2018) [25].
Scaranni et al., (2021) [20]	English	Estimate changes in BP and incidence of AH associated with consumption of UPFs in adults	UPFs: According to Monteiro et al., (2016) [27].
Smiljanec et al., (2020) [21]	English	Investigate association between consumption of UPFs/in natura/minimally processed foods and peripheral/central BP	UPFs: breakfast cereals, packaged bread, flavored yogurt and dairy products, half and half, lactose-free milk, milk alternatives, packaged sliced, processed, and creamed cheese, processed meats, meat alternatives, packaged (instant) soups and noodles, pasta sauces, ready-to-eat frozen dishes, condiments, sweet or salty packaged snacks, ice cream, confectionery, sugar-sweetened beverages, hard liquor). Cheese and dried, cured, or smoked meats were included in the UPFs category as they contain additives such as colors, preservatives, and stabilizers.
Steele et al., (2019) [22]	English	Examine association between participation of UPFs in diet and metabolic syndrome	UPFs: Article followed description by Monteiro et al., (2019) and Martinez Steele et al., (2016) for PFs and UPFs [8,30].

Data reported as mean of AH, arterial hypertension; PFs, processed foods; UPFs, ultra-processed foods; BP, blood pressure.

**Table 2 nutrients-14-01215-t002:** Characteristics of studies selected for present review (*n* = 9).

First Author (Year)	Study Design (Study Period)/Country	Population (Sample Size/Age)	Food Consumption Assessment Method	Dietary Components	Diagnostic Criteria for Hypertension	Energy Contribution of PFs/UPFs (%)	Statistical Analysis Association between Food Processing and Hypertension
Conceição et al., (2018) [19]	Cross-sectional (2014–2015) Brazil	64 adults 25–57 years	One-day 24 hR/ NOVA classification (Monteiro, 2010)	PFs UPFs	Measurement of BP using digital meter according to 6th Brazilian Arterial Hypertension Guidelines (2010).	PFs: 6.5% UPFs: 7.7%	Student’s *t*-test No significant difference in mean SBP or DBP in comparison of individuals based on consumption of food groups (*p* > 0.05)
Martinez-Peres et al., (2021) [24]	Transversal (2020) Spain	5636 adults and older people 55–75 years (mean age: 65 years)	Semi-quantitative FFQ with 143 items (validated *)/NOVA classification (Monteiro, 2010)	UPFs	Use of anti-hypertensive agent and BP equal to or higher than 130/85 mmHg.	UPF: 7.9% **	Linear regression. No significant association between consumption of UPFs and SBP and DBP in adjusted models (β = −0.17 mmHg; CI = −0.5, 0.16; *p* = −0.08 e β = 0.08 mmHg; CI = −0.1, 0.26; *p* = 0.383, respectively).
Mendonça et al., (2017) [1]	Cohort (1999–2015) Spain	14790 middle-aged adults	Self-administered semi-quantitative FFQ with 136 items (validated *)/NOVA (servings/day and caloric contribution)	UPFs	Self-declared medical diagnosis.	UPFs: 2.1 to 5 servings/day ***	Cox regression Positive association between consumption of UPFs and AH. Highest tercile of consumption of UPFs had greater risk of developing AH compared to lowest tercile (HR adjusted by multivariable analysis = 1.21 [95% CI: 1.06–137]).
Monge et al., (2021) [23]	Cohort (2006–2010) Mexico	64 934 women 41.7 (SD: 7.2) years	Semi-quantitative FFQ with 140 items (validated *)/NOVA (caloric contribution)	UPFs	Self-declared medical diagnosis or use of antihypertensive.	UPFs: Total—29.8% (SD: 9.4) Liquid—6.4% (SD: 4.8) Solid—23.4% (SD: 8.9)	Poisson regression Total consumption of UPFs and consumption of solid UPFs not associated with AH (IRR = 0.96, 95% CI: 0.79, 1.16; IRR = 0.91, 95% CI: 0.82, 1.01, respectively). Ultra-processed beverages and processed meats associated with increase in incidence of AH (IRR = 1.32, 95% CI: 1.10, 1.65; IRR = 1.17, 95% CI: 1.01, 1.36, respectively).
Nardocci et al., (2020) [3]	Cross-sectional (2015) Canada	13,608 adults ≥ 19 years	24 hR/NOVA classification (Monteiro, 2010), caloric contribution	UPFs	Self-declared AH—answer to question on long-term health conditions diagnosed by healthcare provider: “Do you have diabetes/high blood pressure?”	UPFs: 47%	Linear regression UPFs significantly associated with greater likelihood of developing AH. In adjusted models, 10 percentage point increase in relative energy from UPFs associated with 9% increase in likelihood of AH. Adults in highest tercile of consumption of UPFs 60% more likely to have AH (OR = 1.60, 95% CI: 1.26–2.03) compared to those in lower terciles. Odds ratio used for 10% increase in relative intake of UPFs (% of total energy intake)
Rezende-Alves et al., (2020) [2]	Cohort (2016–2018) Brazil	1221 Adults (mean age: 35.2 years)	FFQ (validated *)/ NOVA classification (caloric contribution)	PFs UPFs	Self-declared medical diagnosis or use of antihypertensive or self-declared high BP (≥130/80 mmHg) according to recent cutoff points proposed by ACC/AHA.	PFs: 9.9% (SD: 5.8) UPFs: 25.8% (SD: 11)	Poisson regression Highest quintile of consumption of UPFs had increased risk of AH (RR: 1.35; 95% CI: 1.01, 1.81). When alcohol intake was excluded from caloric percentage of UPFs, greater consumption of these foods remained independently associated with increase in incidence of AH (RR: 1.35; 95% CI: 1.01, 1.82). No association identified between PFs and AH.
Scaranni et al., (2021) [20]	Cohort (2008–2010) Brazil	8171 adults and older people 35–74 years (mean: 49 years)	FFQ with 114 items (validated *)/ NOVA (caloric contribution)	UPFs	Measurement of BP (SBP ≥ 140 mmHg or DBP ≥ 90 mmHg) and use of anti-hypertensive in previous two weeks.	UPFs: 25.2% (14.5–35.4%)	Mixed-effects linear regression to evaluate changes in BP and logistic regression to evaluated incidence of AH Greater consumption of UPFs associated with 23% greater risk of developing AH (OR = 1.23, 95% CI: 1.06, 1.44). No association between consumption of UPFs and changes in BP (mean SBP and DBP increased over time and varied slightly with consumption of UPFs).
Smiljanec et al., (2020) [21]	Cross-sectional USA	40 adults 18–45 years	Three-day food record/ NOVA classification (Monteiro, 2010)	UPFs	BP measured by outpatient monitoring. Central and peripheral BP measured by SBP, DBP, MBP, PP and aortic pressure. Monitoring outside clinic followed recommendations of Screening for high blood pressure in adults: U.S. Preventive Services Task Force recommendation statement (2015).	UPFs: 50.0 ± 2.4%	Multiple linear regression Positive association between UPFs and general and diurnal SBP (B = 0.25, 95% CI: 0.03, 0.46, *p* = 0.029; B = 0.32, 95% CI: 0.09, 0.56, *p* = 0.008, respectively), diurnal DBP (B = 0.18, 95% CI: 0.01, 0.36, *p* = 0.049) and diurnal peripheral PP (B = 0.22, 95% CI: 0.03, 0.41, *p* = 0.027). After adjustments, UPFs positively associated with SBP (1% increase in consumption of UPFs associated with 0.25 mmHg and 0.32 mmHg increase in general and diurnal SBP, respectively), peripheral and central DBP. No significant association between consumption of UPFs and BP in men, but tendency toward positive association between UPFs and BP. 95% CI and *p* < 0.05 used.
Steele et al., (2019) [22]	Cross-sectional (2009–2014) USA	6385 adults ≥ 20 years and older people	Two-day R24/NOVA classification (Monteiro, 2010) (caloric contribution)	UPFs	Measurement of BP (SBP ≥ 130 mmHg and/or DBP ≥ 85 mmHg based on Centers for Disease Control and Prevention 2009–2010; 2011–2012; 2013–2014) or use of antihypertensive.	UPFs: 55.5%	Poisson regression Significant association between consumption quintiles of UPFs and increase in BP (PR = 1.19; 95% CI: 1.03, 1.38) in adjusted multivariate models.

Data expressed as mean ± standard deviation (SD); CI, confidence interval; 24 hR, 24-h recall; AH, arterial hypertension; FFQ, food frequency questionnaire; PFs, processed foods; UPFs, ultra-processed foods; BP, blood pressure; SBP, systolic blood pressure; DBP, diastolic blood pressure; MBP, mean blood pressure; PP, pulse pressure. * FFQ validated for population analyzed but not validated for analysis of food intake according to degree of processing. ** The percentage indicates mean consumption of foods and beverages in UPFs group over total intake in grams per day. *** Article did not provide energy contribution of UPFs in percentage.

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
