# Peer review of "A Systematic Review on Processed/Ultra-Processed Foods and Arterial Hypertension in Adults and Older People"

_nutrients, 2022, doi:10.3390/nu14061215_

Round 1

Reviewer 1 Report

  1. This study resulting evidence supports UPFs are associated with a greater risk of developing AHin the adult population and older people. The evidence underscores the need to investigate the eating habits of populations due to the increase in the consumption of unhealthy foods, which can have negative health consequences. Such knowledge could assist in the adoption of measures that have a positive impact on the transformation of the health scenario in the long term.
  2. Try to provide some of biochemical indexes from the cohort and cross-sectional studies?
  3. The format of references is not uniform. At the same time, more new literatures need to be cited.

Reviewer 2 Report

The systematic review, about the possible role of processed/ultra-processed foods in arterial hypertension in adults and older people, submitted by the authors touch a very interesting and important issue.

The paper is well written and is very readable.

However there are some aspect to be revised and improved.

Minor revisions: there are several typos through the text (mainly spaces, missing between words). Page 2, line 70; page 11, lines 183-191; page 12, 252-267; page 14, line 338.

Major revisions:

1) Did the authors follow the PICOS in designing their review? There are no mention in the text about this. I believe this is needed to increase the quality and the clarity of the study design. Please, add PICOS specification in the text and in form of table as supplementary materials.

2) Page 2:  the authors wrote that they used the PRISMA guidelines to conduct their systematic review. Which version of PRISMA guidelines they used? The PRISMA statement was updated in 2020. Please, insert the detail and the reference.

3) In the discussion paragraph, the authors stated "in the present review, four studies used a FFQ without specific validation for the analysis of food intake according to the degree of processing". Please add information about validated or non validated tools in the table 2, under the column "Food consumption assessment method".

Round 2

Reviewer 2 Report

I thank the authors for editing the manuscript in accordance with the proposed revisions.

In my opinion the article has been improved and can be published.